# Therapy Resistance of Glioblastoma in Relation to the Subventricular Zone: What Is the Role of Radiotherapy?

**DOI:** 10.3390/cancers15061677

**Published:** 2023-03-09

**Authors:** Ekin Ermiş, Alexander Althaus, Marcela Blatti, Emre Uysal, Dominic Leiser, Shokoufe Norouzi, Elena Riggenbach, Hossein Hemmatazad, Uzeyir Ahmadli, Franca Wagner

**Affiliations:** 1Department of Radiation Oncology, Inselspital, Bern University Hospital, University of Bern, 3010 Bern, Switzerland; 2Center for Proton Therapy, Paul Scherrer Institute, 5232 Villigen, Switzerland; 3Department of Diagnostic and Interventional Neuroradiology, Inselspital, Bern University Hospital, University of Bern, 3010 Bern, Switzerland

**Keywords:** glioblastoma microenvironment, therapeutic resistance, subventricular zone, radiotherapy

## Abstract

**Simple Summary:**

Glioblastoma is a highly aggressive form of brain tumors characterized by rapid growth. The subventricular zone (SVZ) is an important region in the brain for processes such as neurogenesis, learning and memory, and brain repair after injury. Some studies have shown that glioblastoma stem cells can originate from the SVZ and have been reported to be responsible for the recurrent nature of the disease and for the failure of conventional treatments. Irradiation of these regions could improve patient outcomes. In our study, we investigated the impact of targeting the ipsilateral SVZ and preserving the contralateral SVZ during radiation treatment on patient outcomes.

**Abstract:**

Glioblastoma is a highly heterogeneous primary malignant brain tumor with marked inter-/intratumoral diversity and a poor prognosis. It may contain a population of neural stem cells (NSC) and glioblastoma stem cells that have the capacity for migration, self-renewal and differentiation. While both may contribute to resistance to therapy, NSCs may also play a role in brain tissue repair. The subventricular zone (SVZ) is the main reservoir of NSCs. This study investigated the impact of bilateral SVZ radiation doses on patient outcomes. We included 147 patients. SVZs were delineated and the dose administered was extracted from dose–volume histograms. Tumors were classified based on their spatial relationship to the SVZ. The dose and outcome correlations were analyzed using the Kaplan–Meier and Cox proportional hazards regression methods. Median progression-free survival (PFS) was 7 months (range: 4–11 months) and median overall survival (OS) was 14 months (range: 9–23 months). Patients with an ipsilateral SVZ who received ≥50 Gy showed significantly better PFS (8 versus 6 months; *p* < 0.001) and OS (16 versus 11 months; *p* < 0.001). Furthermore, lower doses (<32 Gy) to the contralateral SVZ were associated with improved PFS (8 versus 6 months; *p* = 0.030) and OS (15 versus 11 months; *p* = 0.001). Targeting the potential tumorigenic cells in the ipsilateral SVZ while sparing contralateral NSCs correlated with an improved outcome. Further studies should address the optimization of dose distribution with modern radiotherapy techniques for the areas surrounding infiltrated and healthy SVZs.

## 1. Introduction

Glioblastoma is the most aggressive primary brain tumor and accounts for approximately 50% of all malignancies of the primary central nervous system [1]. Despite recent advances in understanding its biology and treatment options, including surgery, radiation therapy, systemic treatment (chemotherapy and targeted therapy), and tumor-treating fields, the prognosis remains very poor, with a 5-year survival rate of less than 10% [2]. The underlying reason for this dismal outcome may be multifactorial. Glioblastomas demonstrate highly heterogeneous cell composition and may contain a population of both neural stem cells (NSC) and glioblastoma stem cells (GSC) [3,4]. NSCs are a subtype of progenitor cells in the nervous system, capable of driving neurogenesis and gliogenesis [3,5]. The two main adult neurogenic niches, the subventricular zone (SVZ) of the lateral ventricle and the hippocampal dentate gyrus, contain adult NSCs (Figure 1). NSCs have the ability to self-renew, proliferate, regenerate and migrate. All of these functions are related to repairing the nervous system and maintaining or restoring function, as well as initiating tumors and resistance to current therapies [6,7,8]. The data suggest that glioblastoma-initiating cells display NSC properties, indicating that NSCs could play a major role in tumor development [7,9]. Consequently, regions of neurogenic niches that harbor NSCs have been intensively studied in recent years in an effort to understand glioblastoma pathogenesis, the efficacy of the available therapies and resistance to therapy. The SVZ, an area with a thickness of 3–5 mm, lining the lateral ventricles, is a widely accepted neurogenic niche in adult neurogenesis, whereas the role of the dentate gyrus is still uncertain [10,11]. Earlier observations indicate that the location of glioblastoma and its relationship to the SVZ influence patient outcomes and may negatively impact the contact of the tumor with the SVZ [12,13]. Furthermore, it has been suggested that SVZ-derived tumors may play a role in the therapy resistance of glioblastoma [12,14]. These findings have inspired further clinical studies, focusing on SVZ as a radiation target. The results were controversial, with some showing an additional benefit of higher radiation doses to the SVZ, whereas others did not [15,16,17,18,19,20]. Moreover, irradiation of the SVZ, which impairs its ability to repair normal brain tissue, may have an adverse impact on patient outcomes. This has been shown in several studies that have examined the prognostic value of cognitive deterioration for survival in patients with glioma [21,22]. Supporting this hypothesis, some authors reported an inversely correlated outcome when a healthy contralateral SVZ received higher radiation doses [23,24].

In this study, we investigated the impact of targeting the ipsilateral SVZ (iSVZ) and preserving the contralateral SVZ (cSVZ) during radiation treatment on patient outcomes.

## 2. Materials and Methods

### 2.1. Patient Selection

Patients with histopathologically diagnosed primary supratentorial glioblastoma who underwent and completed adjuvant radiotherapy with or without chemotherapy between December 2013 and December 2018 were retrospectively identified. Patients were excluded if they had glioma histology other than glioblastoma, were younger than 18 years, had a history of previous radiotherapy, or insufficient radiological/dosimetric data. The cantonal research ethics board gave its approval for this investigation.

### 2.2. Treatment and Data Collection

All patients underwent surgical interventions (gross total resection (GTR), subtotal resection (STR) and biopsy) and received radiotherapy according to a standard protocol, with a median dose of 60 Gy (range: 40.05–60 Gy) at a median dose of 2 Gy per fraction (range: 2–2.67 Gy), according to the volumetric modulated arc therapy technique. Hypofractionated radiotherapy was mainly performed in elderly patients (>70 years). For various dose/fractionation schemes, the biologically effective doses were retrospectively recalculated using the linear quadratic model with a dose–response constant (α/β ratio) of 2 Gy [25]. The gross tumor volume (GTV) was defined as the volume of the resection cavity plus any additional contrast-enhanced lesion visible on the post-contrast 3D T1 magnetization-prepared rapid gradient echo magnetic resonance imaging (MRI) sequence. The clinical target volume was created with a 1.5–2 cm margin around the GTV and was adapted to encompass peritumoral edema. Treatment plans were generated using Eclipse (Varian Medical systems, Palo Alto, CA, USA) software.

Clinical data were collected retrospectively from institutional medical records and included patient age, sex, Karnofsky Performance Scores (KPS), O^6^-methylguanine methyltransferase (MGMT) gene methylation statuses, isocitrate dehydrogenase (IDH)1 statuses, the extent of resection (GTR, STR, biopsy), steroid usage, radiological imaging and treatment and clinical outcome data.

### 2.3. SVZ MRI Characteristics, Delineation and Dosimetric Data

Based on the spatial relationship between the tumor and SVZ/cortex, glioblastomas were classified into four groups, as previously described [12] (Figure 2). The iSVZ and cSVZ were delineated on axial acquisition planning computed tomography images according to Barani et al. [3]. The SVZ was described as a 5 mm expansion along the lateral ventricles (Figure 3). SVZ delineation was performed retrospectively, and no dose constraints were given during the plan optimization process. Radiation doses (Dmax and Dmean) received by the iSVZ and cSVZ were obtained from the treatment planning software.

### 2.4. Outcome Measures

Clinical data including radiological images were evaluated before radiotherapy, 4 weeks after radiotherapy and then at 3-month intervals. The endpoints of the study were overall survival (OS) and progression-free survival (PFS). OS was defined as the period from the date of diagnosis to the date of death. PFS was calculated as the interval from the date of diagnosis to the date of radiological recurrence or death. Two experienced neuroradiologists identified radiological recurrence by evaluating the longitudinal follow-up MRI scans, taking into account the Response Assessment in Neuro-Oncology (RANO) criteria [26]. Our in-house standardized MRI glioblastoma protocol was used for all the patients. All sequences were acquired on one of our 1.5 T MRI scanners (Siemens Avanto and Siemens, Aera, Siemens, Erlangen/Germany). The protocol included all patients. The following four standard MRI sequences that constitute the neuro-oncological protocol according the RANO criteria were used: a 3D T2w SPACE (acquired in sagittal direction, TE 380 ms, TR 3000 ms, FOV 256 × 256 mm^2^, FA 120°, isotropic voxel size of 1 mm × 1 mm × 1 mm), an axial FLAIR acquired as a 2D T2w FLAIR in the axial direction (TE 80 ms, TR 8000 ms, FOV 256 × 256 mm^2^, FA 120°, anisotropic voxel size of 1 mm × 1 mm × 3 mm), a native 3D T1w MPR (acquired in sagittal direction, TE 2.67 ms, TR 1580 ms, FOV 256 × 256 mm^2^, FA 8°, isotropic voxel size of 1 mm × 1 mm × 1 mm), a 3D T1w MPRAGE with gadolinium contrast enhancement (sagittal acquisition, TE 4.57 ms, TR 2070 ms, FOV 256 × 256 mm^2^, FA 15°, isotropic voxel size of 1 mm × 1 mm × 1 mm).

### 2.5. Statistical Analysis

Statistical analyses were performed using the IBM SPSS Statistics for Windows software, version 26 (IBM Corp., Armonk, NY, USA). Categorical variables were presented as numbers (percentage). Continuous variables were presented as medians (range). The compliance of the numerical values to the normal distribution was examined using histograms, in addition to Kolmogorov–Smirnov and Shapiro–Wilk tests. Since the quantitative variables did not display normal distributions, two independent groups were compared using the Mann–Whitney U-test. The chi-square test was used to compare the proportions in different groups. Time-to-event data were derived using Kaplan–Meier methodology. Factors that affect outcomes were analyzed using the log-rank test. Statistically significant factors in log-rank tests were evaluated by multivariate Cox regression analysis. An overall *p*-value of < 0.05 was considered to be statistically significant.

## 3. Results

### 3.1. Study Characteristics

A total of 182 successive medical records were retrieved for this study; 147 patients (33% female, 67% male) met the inclusion criteria and were included in the analysis. The median age was 64 years (range: 18–89 years) and the median follow-up period was 14 months (range: 1–86 months). The patient, tumor and treatment characteristics of the whole cohort and subgroups treated with conventional (*n* = 95, 65%) and hypo-fractionation (*n* = 52, 35%) radiotherapy schemes are displayed in Table 1. Twenty-six percent of the patients had multiple glioblastoma (multifocal 5%, multicentric 21%). A total of 46% of the patients had GTR. All patients had completed the course of prescribed radiotherapy and definitive radiotherapy (for inoperable patients) was performed in 22% of the patients. In-field recurrences were observed in 68% of the patients and out-of-field recurrences in 14%. Concomitant and adjuvant chemotherapy was administered to 85% of the patients. Seventy-one percent of the patients had SVZ type I and type II tumors (tumor in direct contact with the SVZ). Overall, 92% of the patients had died by the time of the final analysis.

The incidental mean radiation biologically effective dose BED2 Gy doses on the iSVZ and cSVZ were 85.8 Gy (range: 2.1–123.4 Gy) and 47.2 Gy (1.6–114 Gy), respectively. The iSVZ received a maximum dose of 122.4 Gy (BED2 Gy; range: 46.2–143.7 Gy). The maximum dose to the cSVZ was 82.2 Gy (BED2 Gy; range: 12.4–124.8 Gy).

### 3.2. Prognostic Factors for PFS

The median PFS was 7 months (range: 4–11 months). Table 2 outlines the prognostic factors for PFS in the univariate and multivariable analysis. Of the favorable factors that are significant for PFS in the univariate analysis, KPS > 70 (*p* = 0.028; HR: 1.953 (95% CI 1.074–3.551)), MGMT status (*p* = 0.012; HR: 0.616 (95% CI 0.413–0.899)), Dmax iSVZ (*p* = 0.004; HR: 1.904 (95% CI 1.223–2.965)) and Dmean cSVZ (*p* = 0.034; HR: 1.662 (95% CI 1.039–2.659)) remained statistically significant in the multivariable analysis. No association between SVZ classification and PFS was found.

### 3.3. Prognostic Factors for OS

The median OS was 14 months (range: 9–23 months). The 1-, 2-, 3- and 5-year survival rates were 61.5%, 23.4%, 11.2% and 4.4%, respectively. Table 3 summarizes the prognostic factors for OS from the univariate and multivariable analysis. Of the favorable factors that are significant for OS in the univariate analysis, the multivariable analysis confirmed KPS >70 (*p* = 0.034; HR: 1.679 (95% CI 1.039–2.713)), MGMT methylation (*p* = 0.010; HR: 0.632 (95% CI 0.446–0.897)), Dmax iSVZ (*p* ≤ 0.001; HR: 2.254 [95% CI 1.476–3.442]) and Dmax cSVZ (*p* = 0.002; HR: 2.070 (95% CI 1.296–3.307)) as prognostic factors for OS. No correlation was found between SVZ classification and OS.

### 3.4. Impact of SVZ Dose

The mean and maximum radiation doses to the iSVZ and cSVZ were used to set cut-off values for the Kaplan–Meier analysis. Table 4 depicts the impact of SVZ dose on patient outcomes. For iSVZ, the Dmax of BED2 Gy > 100 Gy (EQD2 > 50 Gy) showed a significant increase in PFS (*p* < 0.001, 8 months (95% CI 6.978–9.022) versus the value recorded at 6 months (95% CI 5.036–6.964)) and in OS (*p* < 0.001, 16 months (95% CI 13.658–18.342) versus 11 months (95% CI 6.885–15.115)). In contrast, for cSVZ, the Dmean BED2 Gy < 64 Gy (EQD2 < 32 Gy) significantly increased PFS (*p* = 0.030, 8 months (95% CI 6.983–9.017) versus the value recorded at 6 months (95% CI 4.907–7.093)) and OS (*p* = 0.001, 15 months (95% CI 12.744–17.256) versus 11 months (95% CI 6.780–15.220)). Additionally, the Dmax of BED2 Gy < 100 Gy (EQD2 < 50 Gy) to the cSVZ led to a significant increase in OS (*p* = 0.006, 15 months (95% CI 12.509–17.491) versus the value recorded at 12 months (95% CI 9.352–14.648)). The corresponding Kaplan–Meier estimates are shown in Figure 4.

## 4. Discussion

Lim et al. developed an MRI-based classification of glioblastoma characterized by the tumor involvement of the SVZ and cortex and demonstrated that tumors that were in contact with the SVZ might be more aggressive and invasive. Since then, the role of the SVZ in relation to prognosis and therapeutic approaches has been widely studied [12]. In the present study, we hypothesized that the radiation dose received by the healthy and infiltrated SVZ would have an impact on the outcome.

The adult SVZ, the largest source of NSCs, and its microenvironment modulate neurogenesis during brain repair. NSC initiate various cascades of intracellular signaling that result in migration cells being directed to the site of damage, such as after a stroke or following demyelination [27]. Ionizing radiation, on the other hand, can dramatically ablate neurogenesis, and specifically targeting NSC may compromise tissue repair [28]. Achanta et al. explored the effect of localized radiation on different cell types in the SVZ of mice and demonstrated that following the delivery of a single dose of 10 Gy, neuroblasts failed to migrate through the irradiated regions of the SVZ and rostral migratory stream route [29]. By contrast, Gonzales et al. preclinically showed that after SVZ irradiation, a small subset of NSCs remained in the SVZ and maintained their ability to be reactivated and respond to a demyelinating lesion by increasing the number of proliferative cells and neuroblasts [30]. The authors concluded that NSC are resistant to SVZ radiation and have the ability to repopulate the neurogenic niche after localized radiation. However, these findings were based on mouse models and translating this information to human models should be performed with caution, due to the anatomical and physiological differences between these species.

Aside from the well-known repair abilities of NSCs, this cell type may also play a role in the origin of glioblastoma. Several research groups have found that glioblastomas contain tumorigenic cells and brain tumor cells associated with NSC [31,32,33]. Recently, Lee et al. presented direct molecular genetic evidence from patients with glioblastoma [34]. Using patient brain tissue, the authors demonstrated that NSCs in the SVZ could be the cells of origin by containing the driver mutations of glioblastoma in patients with their matching tumors. They sampled ipsilateral normal SVZ tissue at a distance from the tumor and observed that glioblastoma driver mutations, such as TERT promoter mutations, were detectable in 56.3% of IDH wild-type patients. Another study focused on the common features between glioma stem cells and NSC in the SVZ and summarized these features, which included nestin expression, high motility, proliferative potential, association with blood vessels, and bilateral communication with constituents of the niche [8].

These findings encouraged us to investigate the tumoricidal effect of administering ipsilateral SVZ irradiation, while protecting the contralateral SVZ, to induce tissue repair of healthy brains after irradiation. Brain tissue repair after ionizing radiation is linked to neurocognitive deficits and neurocognitive decline is known to be a risk in patients with high-grade glioma [21]. In our study, we showed that 32 and 50 Gy were the meaningful cut-off mean and maximum doses, respectively, for the cSVZ, and patients receiving lower radiation doses to the cSVZ had a better PFS and OS. Furthermore, patients who received a mean dose of >40 Gy and a maximum dose of >50Gy to the iSVZ showed an improved outcome. Findings on this topic in the literature are highly controversial. Most of the previous studies demonstrated an outcome benefit with increased SVZ doses [23,34,35,36,37]. However, other research groups showed detrimental effects of irradiating the SVZ with higher doses [15,38,39,40]. In a recent meta-analysis, the authors analyzed the effect of high- versus low-dose irradiation on the SVZ and found statistically significant results (a PFS benefit) at higher doses when administered ipsilaterally. When applied contralaterally, there was no correlation between higher doses and the outcome [41]. According to our results, to improve PFS and OS in patients with glioblastoma, higher radiation doses should be applied to the ipsilateral neurogenic niches to decrease their tumorigenic potentials and modify the tumor microenvironment, resulting in a reduction in the migration of NSC into the tumor. Moreover, higher doses to the contralateral neurogenic niches should be avoided so as not to impair the ability of the SVZ to respond to brain damage, including post-radiation demyelination.

Retrospective studies showed that direct contact of the tumor with the SVZ resulted in a compromised outcome, particularly a reduced OS [42,43,44]. In addition, Hallaert et al. demonstrated that tumors in contact with the SVZ were larger and a greater proportion had an MGMT unmethylated promoter status [42]. In our study, patients with tumors that were in contact with the SVZ showed reduced median PFS (6 months versus 8 months) and OS (14 months versus 17 months), but the difference was not statistically significant. Hypothetically, tumors infiltrating the SVZ have a potential tumorigenic stem cell pool that could promote the proliferation and migration of glioblastoma progenitor cells, thereby enhancing tumor growth and progression [10,45]. However, Kappadakunnel et al. found no evidence of a stem cell-derived genetic signature specific for glioblastoma in contact with the SVZ [46]. Nevertheless, tumors that touch the SVZ could be more centrally located with a greater tendency to infiltrate and pass the midline, hereby associated with a poorer outcome [47,48].

Our study has several weaknesses due to its retrospective design. Selection bias is possibly the most important limitation. Furthermore, SVZ delineation was performed retrospectively, and no dose constraints were defined during the plan optimization process. Accordingly, the radiotherapy dose cut-offs given for the SVZ are incidental doses. Furthermore, tumor location and relation to the SVZ dose were not appraised. Predictably, tumor sites near or bordering the SVZ would increase the doses received by the SVZ. Additionally, neurocognitive performance was not assessed before or after irradiation, as it was not a routine procedure. Moreover, our glioblastoma cohort included patients with IDH mutations, according to the previous WHO classification, which could have interfered with the results. Since the fifth edition of the WHO classification of tumors of the central nervous system (WHO CNS5), published in 2021, IDH-mutant glioblastoma is now referred to as IDH-mutant astrocytoma. Due to this refinement of the classification, we did not investigate the impact of IDH status on patient outcomes. Nevertheless, we have presented the results of a relatively large and homogeneously treated glioblastoma patient cohort. Our results should be validated in a prospective study. 

## 5. Conclusions

In this cohort of patients with glioblastoma, targeting the potential ipsilateral tumorigenic cells in the SVZ while sparing the contralateral NSC correlated with an improved outcome. With modern radiotherapy techniques and delivery systems, it is possible to design a prospective concept with selective targeting of infiltrated and healthy NSCs. Further studies in this field will clearly be crucial for gaining a deeper insight into the use of radiotherapy, not only to improve treatment outcomes, but also to prevent adverse effects, such as neurocognitive decline.

## Figures and Tables

**Figure 1 cancers-15-01677-f001:**
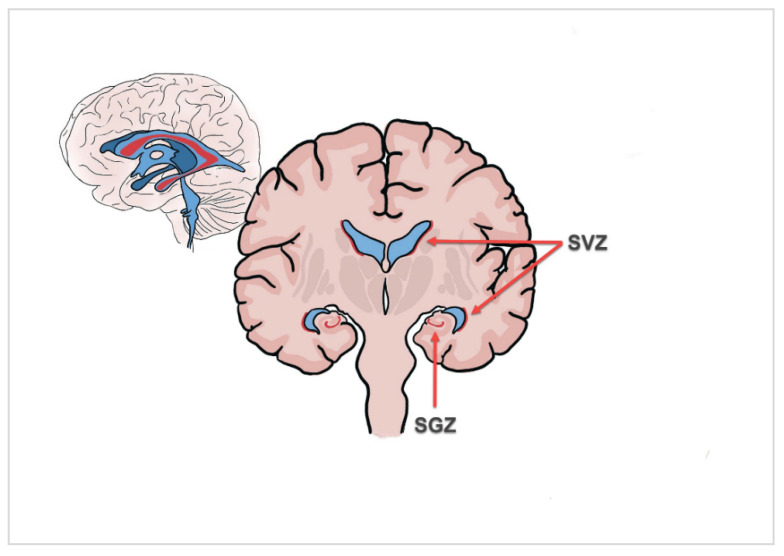
Illustration of the two main adult neurogenic niches, the subventricular zone (SVZ) of the lateral ventricle and the hippocampal dentate gyrus (subgranular zone; SGZ).

**Figure 2 cancers-15-01677-f002:**
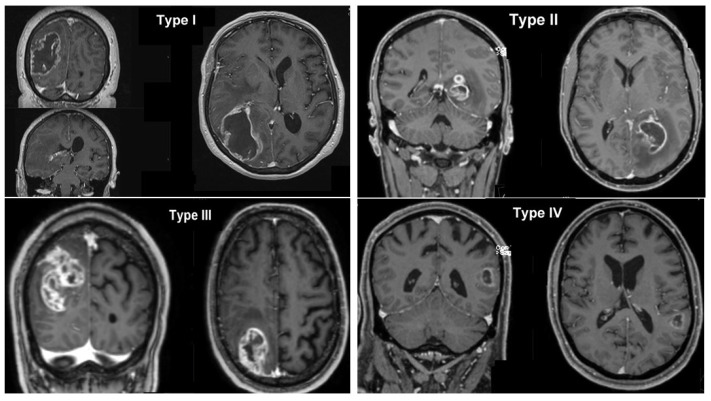
Glioblastoma grouping based on the spatial relationship between the tumor and the SVZ and/or cortex. Type I: Infiltration of both the SVZ and cortex. Type II: Infiltration of the SVZ only. Type III: Infiltration of the cortex only. Type IV: No relation with either SVZ or cortex.

**Figure 3 cancers-15-01677-f003:**
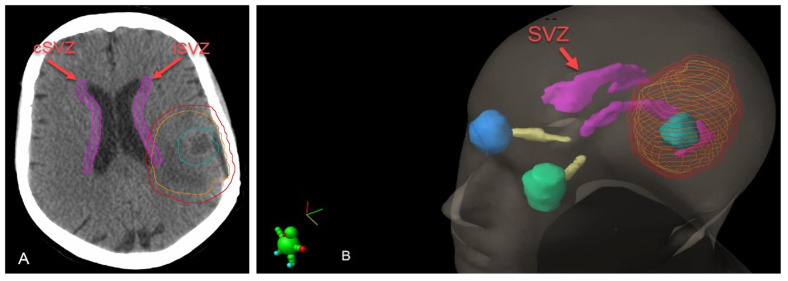
(**A**) Segmentation of SVZ (pink), GTV (blue), CTV (orange) and PTV (red) on the axial CT image. (**B**) Three-dimensional reconstruction of the segmentations. SVZ; subventricular zone. iSVZ; ipsilateral subventricular zone. cSVZ; contralateral subventricular zone.

**Figure 4 cancers-15-01677-f004:**
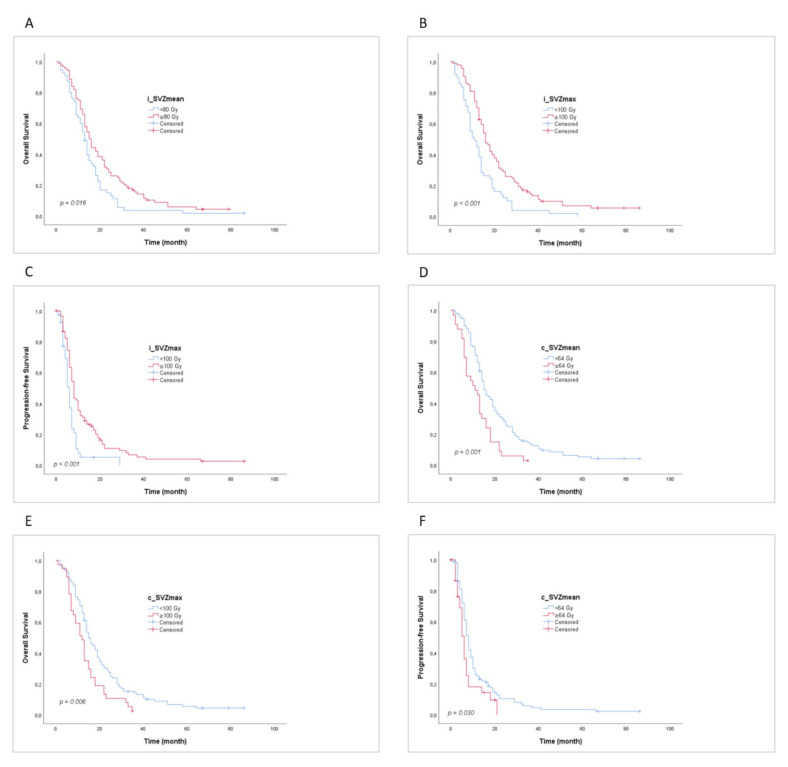
Kaplan–Meier estimates for progression-free survival (PFS) and overall survival (OS). (**A**) OS curve showing patient survival over time with respect to the mean ipsilateral subventricular zone (iSVZ) dose. (**B**) OS curve showing patient survival over time with respect to the max. iSVZ dose. (**C**) PFS curve showing patient survival over time with respect to the max iSVZ dose. (**D**) OS curve showing patient survival over time with respect to the mean contralateral subventricular zone (cSVZ) dose. (**E**) OS curve showing patient survival over time with respect to the max. cSVZ dose. (**F**) PFS curve showing patient survival over time with respect to the mean cSVZ dose.

**Table 1 cancers-15-01677-t001:** Patient, tumor and treatment characteristics.

Characteristics	Median (Range)/*n* (%)
*n* = 147	
Age, years	64 (15–89)
Sex	
Female	48 (32.7)
Male	99 (67.3)
Karnofsky Performance Score	
90–100%	89 (60.6)
80%	34 (23.1)
≤70%	24 (16.3)
Multiple glioblastoma	
Multifocal	7 (4.8)
Multicentric	31 (21.1)
Glioblastoma classification	
Type I	100 (68)
Type II	4 (2.7)
Type III	41 (27.9)
Type IV	2 (1.4)
O^6^-methylguanine methyltransferase	
Methylated	67 (45.6)
Non-methylated	79 (53.7)
NA	1 (0.7)
Isocitrate dehydrogenase-1 gene	
Wild-type	133 (90.5)
Mutant	11 (7.5)
NA	3 (2)
Extent of resection	
Gross total resection	67 (45.6)
Subtotal resection	46 (31.3)
Biopsy	34 (23.1)
Radiotherapy	
Definitive	33 (22.4)
Adjuvant	114 (77.6)
Radiotherapy dose/fractionation	
Conventional (2 Gy)	95 (64.6)
Hypo-fractionation (2.67 Gy)	52 (34.7)
Steroid usage	43 (29.3)
Dmean iSVZ, Gy BED2 Gy 80 Gy = EQD2 40 Gy	85.8 (2.10–123.40)
≥80 Gy	92 (62.6)
<80 Gy	55 (37.4)
Dmax iSVZ, Gy BED2 Gy 100 Gy = EQD2 50 Gy	122.4 (46.2–143.7)
≥100 Gy	96 (65.3)
<100 Gy	51 (34.7)
Dmean cSVZ, Gy BED2 Gy 64 Gy = EQD2 32 Gy	47.2 (1.63–114)
<64 Gy	111 (75.5)
≥64 Gy	36 (24.5)
Dmax cSVZ, Gy BED2 100 Gy = EQD2 50 Gy	82.2 (12.4–124.8)
≥100 Gy	42 (28.6)
<100 Gy	105 (71.4)
Chemotherapy	
Concomitant + adjuvant	109 (74.1)
Only concomitant	15 (10.2)
Only adjuvant	5 (3.4)
Chemotherapy regimen	
Temozolomid	126 (85.7)
Bevacizumab	3 (2)
NA	18 (12.2)
Recurrence	
In field	100 (68)
Out of field	21 (14.3)
NA	12 (8.2)
Salvage therapy	
Radiotherapy	8 (5.4)
Surgery	31 (21.1)
Chemotherapy	58 (39.5)
Best supportive care	36 (24.5)
NA	14 (9.5)
Response assessment in neuro-oncology (RANO)	
Complete response	1 (0.7)
Partial response	9 (6.1)
Stable disease	43 (29.3)
Progression	81 (55.1)
NA	13 (8.8)
Final outcome	
Alive	8 (5.4)
Dead	135 (91.8)
NA	4 (2.7)

iSVZ; ipsilateral subventricular zone, cSVZ; contralateral subventricular zone, NA; not available, BED; radiation biologically effective dose.

**Table 2 cancers-15-01677-t002:** Prognostic factors for PFS.

	Univariate Analysis			Multivariable Analysis		
Factors	*p* Value	Hazards Ratio (HR)	95% Confidence Interval	*p* Value	Hazards Ratio (HR)	95% Confidence Interval
Karnofsky Performance Score (ref. = >70%)	<0.001	2.834	1.624–4.945	0.028	1.953	1.074–3.551
O^6^-methylguanine methyltransferase (ref. = non-methylated)	0.008	0.599	0.410–0.874	0.012	0.616	0.413–0.899
Extent of resection (ref. = biopsy)	0.010	1.832	1.154–2.909			
Chemotherapy (ref. = yes)	0.024	1.925	1.090–3.400			
Dmean iSVZ BED2 Gy (ref. = ≥80 Gy)	0.092	1.380	0.949–2.007			
Dmax iSVZ BED2 Gy (ref. = ≥100 Gy)	0.001	2.000	1.337–2.992	0.004	1.904	1.223–2.965
Dmean cSVZ BED2 Gy (ref. <64 Gy)	0.042	1.580	1.016–2.457	0.034	1.662	1.039–2.659

iSVZ; ipsilateral subventricular zone. cSVZ; contralateral subventricular zone.

**Table 3 cancers-15-01677-t003:** Prognostic factors for OS.

	Univariate Analysis			Multivariable Analysis		
	*p* Value	Hazards Ratio (HR)	95% Confidence Interval	*p* Value	Hazards Ratio (HR)	95% Confidence Interval
Karnofsky Performance Score (ref. = >70%)	<0.001	2.426	1.543–3.814	0.034	1.679	1.039–2.713
Age (ref. = <70 years)	0.032	1.513	1.037–2.207			
O^6^-methylguanine methyl-transferase (ref. = non-methylated)	0.005	0.609	0.430–0.863	0.010	0.632	0.446–0.897
Chemotherapy (ref. = yes)	0.005	2.079	1.255–3.445			
Dmean iSVZ BED2 Gy (ref. = ≥80 Gy)	0.020	1.515	1.067–2.153			
Dmax iSVZ BED2 Gy (ref. = ≥100 Gy)	0.001	1.877	1.314–2.683	<0.001	2.254	1.476–3.442
Dmean cSVZ BED2 Gy (ref. <64 Gy)	0.001	1.957	1.303–2.940			
Dmax cSVZ BED2 Gy (ref. = <100 Gy)	0.008	1.688	1.143–2.492	0.002	2.070	1.296–3.307

iSVZ; ipsilateral subventricular zone. cSVZ; contralateral subventricular zone.

**Table 4 cancers-15-01677-t004:** SVZ dose distribution and outcome.

SVZ Dose	Subgroup	Median PFS (mo)	95% Confidence Interval	*p* Value	Median OS (mo)	95% Confidence Interval	*p* Value
Dmean iSVZ	≥80 Gy (BED2 Gy) ≥40 Gy (EQD2)	8	6.846–9.154	0.072	15	12.702–17.298	0.016
	<80 Gy (BED2 Gy) <40 Gy (EQD2)	6	4.788–7.212		13	11.052–14.948	
Dmax iSVZ	≥100 Gy (BED2 Gy) ≥50 Gy (EQD2)	8	6.978–9.022	<0.001	16	13.658–18.342	<0.001
	<100 Gy (BED2 Gy) <50 Gy	6	5.036–6.964		11	6.885–15.115	
Dmean cSVZ	≥64 Gy (BED2 Gy) ≥32 Gy (EQD2)	6	4.907–7.093	0.030	11	6.780–15.220	0.001
	<64Gy (BED2 Gy) <32 Gy (EQD2)	8	6.983–9.017		15	12.744–17.256	
Dmax cSVZ	≥100 Gy (BED2 Gy) ≥50 Gy (EQD2)	6	5.224–6.776	0.090	12	9.352–14.648	0.006
	<100Gy (BED2 Gy) <50 Gy (EQD2)	8	7.013–8.987		15	12.509–17.491	

OS; overall survival. PFS; progression-free survival. iSVZ; ipsilateral subventricular zone. cSVZ; contralateral subventricular zone.

## Data Availability

The data are kept in the sharepoint provided by the Clinical Trial Unit, documenting the login and user details (https://netz.insel.ch/de/direktionen/lehre-forschung/forschungsdatenbanken/sharepoint/, accessed on 1 March 2022). Changes in the documents are traceable. All identifying data (e.g., names, addresses, date of birth, patient number, etc.) are kept separate from the study data. All digital documents are password-protected.

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
