# Peer review of "Therapy Resistance of Glioblastoma in Relation to the Subventricular Zone: What Is the Role of Radiotherapy?"

_cancers, 2023, doi:10.3390/cancers15061677_

Round 1

Reviewer 1 Report

This is an interesting manuscript which adds to the emerging body of literature on the topic of SVZ and glioma. It is very nearly ready for publication. 

By way of background, the authors summarise describe the current evidence and controversies regarding the SVZ clearly and evenly. The rationale for their study is clear. 

The methodology is clearly described.

Their cohort of patients with GBM treated for 2013-2018 is included.  Exclusion and inclusion criteria are clear. The treatment this cohort received is clearly described and is in keeping with international standard for the time. ( More recently the importance of including T2/FLAIR when considering the GTV and CTV has emerged).  The investigators retrospectively defined a radiologic SVZ according to a well-accepted definition ( 5mm from wall of ventricle) and were thus able to analyse any relationship between dose to ipsilateral and contralateral SVZs.

They found a statistically significant relationship for PFS with Age, KPS, MGMT, D max iSVZ and Dmean SVZ. For OS with Age, KPS, MGMT, Dmax iSVZ and Dmax cSVZ. The discussion and conclusions are appropriate and directly relate to the presented findings. Relevant weaknesses of the data are discussed. 

Some questions from this reviewer. 

1. Line 166. It is not clear what the authors mean by the terminology 'definitive radiotherapy' being received by 22% of patients. Could this be explained a bit more for clarity? 

2. IDH is a dominant PF in Glioblastoma and more important than MGMT ( unless only a pure IDH WT group is being considered).  Thus recommend considering not including the IDH mutant group at all, which is quite small, or including IDH in the UVA/ MVA if statistically valid ( I am not a statistician), or at least addressing this matter in the discussion to explain why not included in the UVA. 

3. It is not really necessary to spell out in sentences all the data that is easily read from the Tables 2-4.  Recommend abbreviating the paragraphs 3.2, 3.3 and 3.4 in this regard.

4. Paragraphs 3.3 - the results are presented as OR, presumably Odds Ratios but the table with the same values, denotes Hazard ratios, HR. Can this be explained or fixed for clarity? 

Author Response

We would like to thank the editor and reviewers for giving the opportunity to revise our manuscript. We carefully considered the comments and remarks offered by the reviewers. Herein, please find the revised paper based (please see the attachment) on those comments and recommendations. In responding to their comments, we believe our manuscript is improved. 

Point 1, Line 166: definitive radiotherapy is administered for 22% of the patients where an operation cannot be performed. Term is explained with adding " for inoperable patients" in bracets.

Point 2: Our glioblastoma cohort included patients with IDH-mutation, according to the previous WHO classification. Since the fifth edition of the WHO classification of tumors of the central nervous System (WHO CNS5), published in 2021, IDH-mutant glioblastoma is now referred to as IDH-mutant astrocytoma. Due to this refinement of the classification, we did not investigate the impact of IDH-status on outcome. We added a short text to the last paragraph in the discussion adressing this issue. 

Point 3: The paragraps are abbreviated accordingly.

Point 4: OR are corrected as HR accordingly in the text 3.3

Reviewer 2 Report

The ms by Ermis et al. presents the research on radiotherapy of SVZ in context of prognosis of GBM. The research is well conducted, the cohort is large enough, the analysis clear, and results accordingly presented. The conclusions are supported by the results. The subject of SVZ in GBM is still being explored, therefore the ms adds an important piece of knowledge to the subject. I would just recommend one thorough reading to correct few punctuation, grammar, spelling and styllystic mistakes.

Author Response

We would like to thank the editor and reviewers for giving the opportunity to revise our manuscript. We are grateful for the time and energy you expended on our behalf. Manuscript has been carefully read through one more time and checked by a native English-speaking colleague. 
